# Adherence to the Mediterranean Diet in Spanish Population and Its Relationship with Early Vascular Aging according to Sex and Age: EVA Study

**DOI:** 10.3390/nu12041025

**Published:** 2020-04-08

**Authors:** Marta Gómez Sánchez, Leticia Gómez Sánchez, Maria C Patino-Alonso, Rosario Alonso-Domínguez, Natalia Sánchez-Aguadero, Cristina Lugones-Sánchez, Emiliano Rodríguez Sánchez, Luis García Ortiz, Manuel A Gómez-Marcos

**Affiliations:** 1Primary Care Research Unit of Salamanca (APISAL), Biomedical Research Institute of Salamanca (IBSAL), 37005 Salamanca, Spain; martagmzsnchz@gmail.com (M.G.S.); leticiagmzsnchz@gmail.com (L.G.S.); carpatino@usal.es (M.C.P.-A.); rosa90alonso@hotmail.com (R.A.-D.); natalia.san.ag@gmail.com (N.S.-A.); crislugsa@gmail.com (C.L.-S.); emiliano@usal.es (E.R.S.); lgarciao@usal.es (L.G.O.); 2Department of Statistics, University of Salamanca (USAL), IGA Research Group, 37007 Salamanca, Spain; 3Department of Nursing and Physiotherapy, University of Salamanca, 37007 Salamanca, Spain; 4Health Service of Castilla and Leon (SACyL), 37005 Salamanca, Spain; 5Spanish Network for Preventive Activities and Health Promotion (redIAPP), Gran Via de les Corts Catalanes, 587, 08007 Barcelona, Spain; 6Iberian Network on Arterial Structure, Central Hemodynamics and Neurocognition, 37007 Salamanca, Spain; 7Department of Medicine, University of Salamanca, 37007 Salamanca, Spain; 8Department of Biomedical and Diagnostic Sciences, University of Salamanca, 37005 Salamanca, Spain

**Keywords:** Mediterranean diet, early vascular aging, normal vascular aging, general population

## Abstract

The objective of this study is to analyze the influence of adherence to the Mediterranean diet (MDA) and its components on early vascular aging (EVA) in a Spanish population sample free of cardiovascular disease and to analyze the differences by sex. Methods: We recruited 501 individuals aged 35–75 without cardiovascular disease by random sampling (55.90 ± 14.24 years, 49.70% men). EVA was defined in two steps: Step 1: subjects with vascular damage in carotid arteries or peripheral artery disease were classified as EVA. Step 2: subjects at the percentile of the combined Vascular Aging Index (VAI) were classified; ≥ p90 was considered EVA and < p90 was considered normal vascular aging (NVA), estimated using the following formula (VAI = (log (1.09) × 10 cIMT + log (1.14) cfPWV) × 39.1 + 4.76 by age and sex. Carotid-femoral pulse wave velocity (cfPWV) was measured by SphigmoCor System^®^ and carotid intima-media thickness by Sonosite Micromax^®^ ultrasound and classified thus: values ≥ Percentile 90 were considered EVA and those < Percentile 90 as NVA, with population percentiles analyzed. The principal result variable was assessed using the 14-item MEDAS questionnaire, developed and validated by the PREDIMED group, comprising 12 questions about the frequency of food consumption and two questions regarding the Spanish population’s typical eating habits. Results: MDA was observed by 25% (17% men and 34% women). EVA was present in 17% (29% men and 4% women). The adjusted logistic regression models showed that an increase in MDA decreases the probability of EVA in the global analysis (OR = 0.36; 95% CI: 0.16–0.82). In the analysis by sex, this association was only seen in men (OR = 0.33; 95% CI: 0.12–0.86), but not in women (OR = 0.31; 95% CI: 0.04–2.50). Conclusion: The results of this study suggest that a greater adherence to the Mediterranean diet decreases the probability of presenting EVA. In the analysis by sex, this association applies only to men.

## 1. Introduction

The Mediterranean diet (MD) consists of a high intake of vegetables and legumes, fruit, fish, white meats, pasta with tomatoes, nuts, and olive oil; and a low intake of red meat, dairy products, and commercial pastries. It is currently considered one of the healthiest dietary patterns, with important antioxidant and anti-inflammatory effects [1]. A review by the Cochrane Library [2] showed the benefits of the Mediterranean diet in primary prevention, reducing mortality from ischemic heart disease [3], improving lipid profile, blood pressure, blood glucose, and adiposity; however, the level of evidence was weak to moderate and no benefits in secondary prevention were found [2]. However, the relationship of the Mediterranean diet with early vascular aging (EVA) has not been studied. Only one review analyzed the beneficial effect of polyphenols (found in fruits, vegetables, greens, nuts, and olive oil) on vascular aging [4].

In addition, the interest in understanding the factors that influence EVA, including the MD, as well as the criteria to define it, has grown in recent years. Therefore, the search for a definition of EVA is currently being addressed by numerous authors. Different definitions of EVA have also been published [5,6,7,8] in various papers using the highest cfPWV percentiles in their definitions. Recently Nilsson W et al. [9] published the vascular aging index, integrating carotid artery intima-media thickness (cIMT) and carotid-femoral pulse wave velocity (cfPWV), which reflect arterial stiffness and subclinical atherosclerosis. Finally, the association of adherence to the Mediterranean diet and its components with EVA, as measured with the vascular aging index [9], has not been studied in the general Spanish population.

The objective of this study is therefore to analyze the influence of adherence to the Mediterranean diet and its components on EVA in a sample of the Spanish population free of previous cardiovascular disease and to analyze the differences by sex.

## 2. Materials and Methods 

### 2.1. Study Design

This is a descriptive transversal study of subjects recruited to investigate the association between different risk factors and early vascular aging (EVA study) [10]. It is registered in ClinicalTrials.gov (NCT02623894).

### 2.2. Study Population

The EVA project was carried out in the Primary Care Research Unit (APISAL). The subjects of this study were selected using random sampling with replacement by age groups (35, 45, 55, 65, and 75 years) from an urban population. The sample comprised of 501 subjects, 100 from each age group, half of each sex. Recruitment was carried out between June 2016 and November 2017. The selection of participants by age groups and sex, as well as reasons for exclusion and reference population (43,946) can be seen in the flowchart shown in Appendix A.

A detailed description of the study methodology with inclusion and exclusion criteria and response rate has been published [10,11].

### 2.3. Ethics Approval and Consent to Participate

The study obtained approval on 4/5/2015 by the Salamanca ethics committee on research with medicines. The recommendations specified in the Declaration of Helsinki were followed throughout [12]. All patients signed written informed consent forms prior to participation in the study. 

### 2.4. Variables and Measuring Instruments

Before the study began, two health professionals were trained to perform the measurements and collect the completed questionnaires following a standardized protocol.

#### 2.4.1. Assessment of Carotid Intima-Media Thickness 

The measurement of the cIMT was performed by two researchers of proven reliability, with intraclass and intraobserver correlation coefficient values of 0.974 and 0.897 for interobserver agreement in measurements made on 20 subjects before beginning the study. The device used was Sonosite Micromax^®^ ultrasound device (Sonosite, Inc., Bothell, WA, USA), with a 5–10 MHz multi-frequency high-resolution linear transducer and with Sonocal software, (Washington, USA) which automatically measures the cIMT. The cIMT measurements were performed following the protocol published by our research group [13]. 

#### 2.4.2. The Ankle Arm Index 

The VaSera VS-1500^®^ device (Fukuda Denshi, Tokyo, Japan) was used for this measurement. The presence of vascular lesion was established following the criteria included in clinical practice guidelines [14].

#### 2.4.3. Assessment of Carotid-Femoral Pulse Wave Velocity (cfPWV)

The cIMT was assessed by two researchers of proven reliability in measurements made on 20 subjects before the beginning of the study, with intraclass and intraobserver correlation coefficients of 0.974 and 0.897 for interobserver agreement. The device used was a Sonosite Micromax^®^ ultrasound (Sonosite, Inc., Bothell, WA, USA), with a 5-10 MHz multi-frequency high-resolution linear transducer, which automatically measures the cIMT through the Sonocal software. The cIMT measurements were performed following the protocol published by our research group [15].

#### 2.4.4. Definition of Healthy Vascular Aging, Normal Vascular Aging and Early Vascular Aging

EVA and *healthy vascular aging* (NVA) were defined in two steps: Step 1: subjects with vascular damage in carotid arteries or peripheral artery disease were classified as EVA. Step 2: The population studied was classified by age and sex in percentiles of Vascular Aging Index (VAI) [9], which was estimated with carotid-femoral pulse wave velocity (cfPWV) and intima-media thickness of the carotid artery, using the following formula VAI = (log (1.09) × 10 cIMT + log (1.14) cfPWV) × 39.1 + 4.76. Thus, values ≥ Percentile 90 were considered EVA and those < Percentile 90 as NVA.

#### 2.4.5. Adherence to the Mediterranean Diet 

This was assessed with a 14-item questionnaire used in the PREDIMED study, previously validated in the Spanish population [16]. The questionnaire includes 12 questions about food consumption frequency and two about eating habits, with scores of zero or one. In short, subjects scored one point for each of the following: using olive oil as the main fat for cooking, eating more white meat than red meat or, daily, eating more than 4 tablespoons of olive oil, 2 or more servings of vegetables, 3 or more pieces of fruit, less than 1 serving of red meat, and less than 1 sugary drink. An additional point was given for the weekly consumption of 7 or more glasses of wine, 3 or more servings of legumes, 3 or more servings of fish, 3 or more servings of nuts or dried fruits, 2 or more servings of stir-fry, and less than 2 portions of cake or similar. The total score ranged from 0 to 14 points and MDA was considered for totals of ≥ 9 points [16].

#### 2.4.6. Evaluation of Lifestyles 

Smoking status was measured with a standardized questionnaire (registering whether or not the subject smoked, and if so, the number of cigarettes smoked and smoking index). Subjects who smoked at the time of the assessment or who had quit smoking during the previous year were defined as smokers. The smoking index = number of cigarettes per day divided by years of smoking. Alcohol use was measured with a standardized questionnaire (recording the type and amount of alcohol drunk during a week, measured in gr/week). Drinking was considered free of risk for quantities below 140 g/week for women and 210 g/week for men. Physical activity was assessed with the International Physical Activity Questionnaire—Short Form (IPAQ-SF): The short form (9 items) records activity at four levels of intensity: (1) intense physical activity, such as aerobics, (2) moderate-intense activity, such as leisure cycling, (3) walking, and (4) being sedentary for 7 days. The data were computed in METS/min/week. Subjects performing at least 30 min of moderate activity 5 days a week, or at least 20 min of vigorous or very vigorous activity 3 days per week were considered active [17]. Sedentary hours per week were evaluated by the Marshall questionnaire [18].

#### 2.4.7. Measurement of Cardiovascular Risk Factors 

The measurement of blood pressure, weight, and height was carried out in the Primary Care Research Unit of Salamanca (APISAL) following the recommendations as previously published in the study protocol [10].

Subjects were considered to have hypertension if they were taking antihypertensive drugs or had blood pressure values ≥ 140/90 mmHg; to have diabetes if taking hypoglycaemic agents or with fasting plasma glucose ≥ 126 mg/dl or HbA1c ≥ 6.5%; to have dyslipidaemia if taking lipid-lowering drugs or with fasting total cholesterol ≥ 240 mg/dl, low density lipoprotein cholesterol (LDL-c) ≥ 160 mg/dl, high density lipoprotein cholesterol (HDL-c) ≤ 40 mg/dl in men and ≤ 50 mg/dl in women, or triglycerides ≥ 150 mg/dl. A value of BMI ≥ 30 kg/m^2^ was used to define obesity [14]. 

### 2.5. Statistical Analysis

Data are presented using means ± standard deviations and numbers or percentages according to whether they are continuous or categorical variables. The comparison between NVA and EVA was performed using chi-square tests for percentages and Student’s t for continuous variables. Fifteen logistic regression models were run (one for global adherence to the MDA and one for each of the 14 components), taking the presence of EVA against NVA as a dependent variable (0 = NVA, 1 = EVA) and MD and each of its components as independent variables (1 = Yes, 0 = No). To select the adjustment variables, we carried out a simple binary logistic regression analysis between MDA and lifestyle, cardiovascular risk factors, and drugs, selecting those with values of *p* < 0.05. The categorical variables we used: age (0 = <60 years, 1 = ≥60 years); sex (0 = woman; 1 = man); presence of hypertension, diabetes or obesity (0 = No; 1 = Yes); to be active (0 = No; 1 = Yes); and hypotensive and hypoglycaemic drugs (0 = No; 1 = Yes). The continuous variables employed were: tobacco index, alcohol use, number of sedentary hours per week, and atherogenic index. The logistic regression analysis was carried out globally and by sex. The SPSS Statistics program for Windows, version 25.0 (IBM Corp, Armonk, NY, USA) was used. We consider a value of *p* < 0.05 as a statistical significance limit.

## 3. Results 

### 3.1. Clinical Characteristics and Vascular Aging

Demographic characteristics, lifestyles, cardiovascular risk factors, and use of drugs, globally and by categories of vascular aging, are shown in Table 1. Mean age was 55.90 ± 14.24 years. Compared to EVA subjects, those categorized as NVA had higher percentages of MDA (28% vs. 15%) and adequate alcohol use (92% vs. 82%), and lower percentages of smoking (14% vs. 27%), hypertension (26% vs. 47%), diabetes (5% vs. 18%), and obesity (18% vs. 23%). 

EVA was found in 16.6% of the participants (men: 28.9% and women 4.4%; *p* < 0.001) Figure 1a. The percentage of participants with EVA by age group is reflected in Figure 1b (*p* = 0.350).

### 3.2. Adherence to the Mediterranean Diet and Its Components

Table 2 shows the Mediterranean diet adherence percentages in global terms and for each of the 14 components of the MEDAS questionnaire in subjects with NVA and EVA. In comparison with EVA subjects, those with NVA had a higher percentage of adherence to the Mediterranean diet (23% vs. 2%), ate more vegetables and greens (23% vs. 12%), fruit (43% vs. 28%), fewer items of commercially produced pastry per week (41% vs. 34%), more nuts (30% vs. 17%), white meat (66% vs. 49%) and pasta with sofrito (49% vs. 32%). On the other hand, they adhered less to the wine drinking criterion (14% vs. 31%).

### 3.3. Association between EVA and Adherence to the Mediterranean Diet and Its Components

The adjusted logistic regression models showed that an increase in the MDA decreases the probability of presenting EVA in the global analysis (OR = 0.36; 95% CI 0.16–0.82) (Figure 2). In the analysis by sex, the association is only seen in men (OR = 0.33; 95% CI 0.12–0.86), but not in women (OR = 0.31; 95% CI 0.04–2.50) (Figure 3). When each of the MD components was individually analyzed in the global model, the probability of presenting EVA decreases with the consumption of: ≥ 3 pieces of fruit per day (OR = 0.51; 95% CI 0.28–0.94); ≥ 3 fish/shellfish dishes per week (OR = 0.49; 95% CI 0.27–0.91); < 2 servings of commercial (non-homemade) pastries such as cookies, custards, sweets or cakes per week (OR = 0.51; 95% CI 0.29-0.90); ≥ 3 servings of nuts per week (OR = 0.49; 95% CI 0.25–0.98); preferably eating chicken, turkey or rabbit meat instead of beef, pork, hamburgers or sausages (OR = 0.47; 95% CI 0.27–0.82); and eating cooked vegetables, pasta with tomato (OR = 0.49; 95% CI 0.25–0.98) ≥ 2 times a week (Figure 2). In the analysis by sex, the only component showing a difference was: ≥3 pieces of fruit per day decreases the probability of presenting EVA in men (OR = 0.43; 95% CI 0.21–0.88) (Figure 3).

## 4. Discussion

To our knowledge, this is the first study that analyzes the relationship of EVA with adherence to the Mediterranean diet and each of its components in the Spanish general population between 35 and 75 years of age and with no history of cardiovascular disease. The main findings of this study are that adherence to the Mediterranean diet is greater in women. EVA was found in 16.6% of the participants, with a higher percentage in men. The logistic regression analysis adjusted for possible confounding variables showed that an increase in the MDA decreases the probability of presenting EVA, but in the analysis by sex, the association is only significant in men, probably due to the low number of women presenting EVA since the OR is similar in both sexes.

The average score and the percentage of adherents to the Mediterranean diet were higher in women, in line with other published data [1,19]. 

The prevalence of EVA (16.6%) found in this study is higher than in the data published in the OPTIMO study [7], at 5.7%, and in those of the study carried out by Cunha et al. [6], at 12.5%. However, it should be taken into account that the prevalence found in the previous studies are not comparable given the use of different criteria to define EVA, population distribution by age and sex, and the presence of risk factors, which reflects the need to establish a consensus and a common definition of EVA.

The association between Mediterranean diet adherence and vascular aging found in our research is supported by population studies and intervention trials, which have shown that the Mediterranean diet is related to a lower prevalence of vascular disease, obesity, arthritis, cancer, and age-related cognitive deterioration. Thus, clinical trials comparing the Mediterranean diet with usual diets in both European [20] and Australian populations [21] have shown decreases in SBP, arterial stiffness, and improvement in endothelial function.

When analyzing the different components in our study, the effect varies. Thus, a study conducted in the Spanish population [22] found that lower figures for reactive C protein are associated with the consumption of vegetables, fruit, dairy products, and fish, but not with the consumption of olive oil or nuts, two important components of the Mediterranean diet.

The beneficial effects of the Mediterranean diet on vascular aging are the result of eating fruit, cereals, pulses, vegetables, and using olive oil as the main source of monounsaturated fatty acids [4]. Recent years have therefore seen numerous studies analyzing the relationship and the mechanisms involved between the components of the Mediterranean diet and aging in both animals and humans [23]. The study conducted in the cohort of nonagenarians participating in the Mugello study suggests that daily consumption of olive oil, fruits, and vegetables can protect against the development of endothelial dysfunction through the increase of endothelial progenitor cells and circulating progenitor cells [24]. 

The review carried out by Serino A et al. [4] on the role of polyphenols in aging indicates that polyphenols can reduce inflammation and increase antioxidant capacity, providing protection against atherosclerosis and vascular aging. Polyphenols are found in fruits and vegetables, olive oil, and wine. However, we should not forget that fruit is also rich in other components such as vitamins and fiber, so it is difficult to conclude that polyphenols are solely responsible for improving aging; in the context of fruit intake, it is plausible that fiber is important for the proper functioning of gut bacteria, which are known to metabolize polyphenols into active ingredients. Likewise, the polyphenols in extra virgin olive oil may play an important role in improving vascular aging. However, a study in mice concludes that the consumption of olive oil may not have a protective role in aging or memory, and the beneficial effects may be linked to the improvement in lipid metabolism [25]. Other studies support the hypothesis that the beneficial effects of consuming extra virgin olive oil on health may be mediated by its effects on mesenchymal stem cells. This may explain part of the health effects of olive oil consumption, such as preventing unwanted aging processes [24].

In any case, we should remember that the effect found in this study only applies to 6 of the 14 components measured to assess adherence to MD; the global results must therefore be interpreted with caution and we should bear in mind that the effect of each of the components on EVA may differ. For all these reasons, we consider that prospective studies in the Spanish population are necessary to clarify these uncertainties. In this sense, the Amsterdam Growth and Health Longitudinal Study [26] concluded that promoting the Mediterranean diet in adolescence and early adulthood may constitute an important means of preventing arterial stiffness in adulthood.

These findings suggest that the total benefit of the Mediterranean diet is due to the cumulative or synergistic impact of several foods instead of just one. However, more studies are needed to clarify both the mechanisms involved and the effects of the different components on vascular aging.

The main limitations of this work are: first, the cross-sectional analysis does not allow inference of causality. Secondly, the results of this study refer only to the Spanish urban population and our results may not be generalizable to other races/ethnicities. Thirdly, the assessment of food intake was carried out via subject self-report through validated questionnaires, thus making it a subjective measurement. Finally, the prevalence of CVRF in this study was lower than that in other studies with Caucasian populations.

## 5. Conclusions

The results of this study suggest that a greater adherence to the Mediterranean diet decreases the probability of presenting EVA. In the analysis by sex, this association only applies to men.

## Figures and Tables

**Figure 1 nutrients-12-01025-f001:**
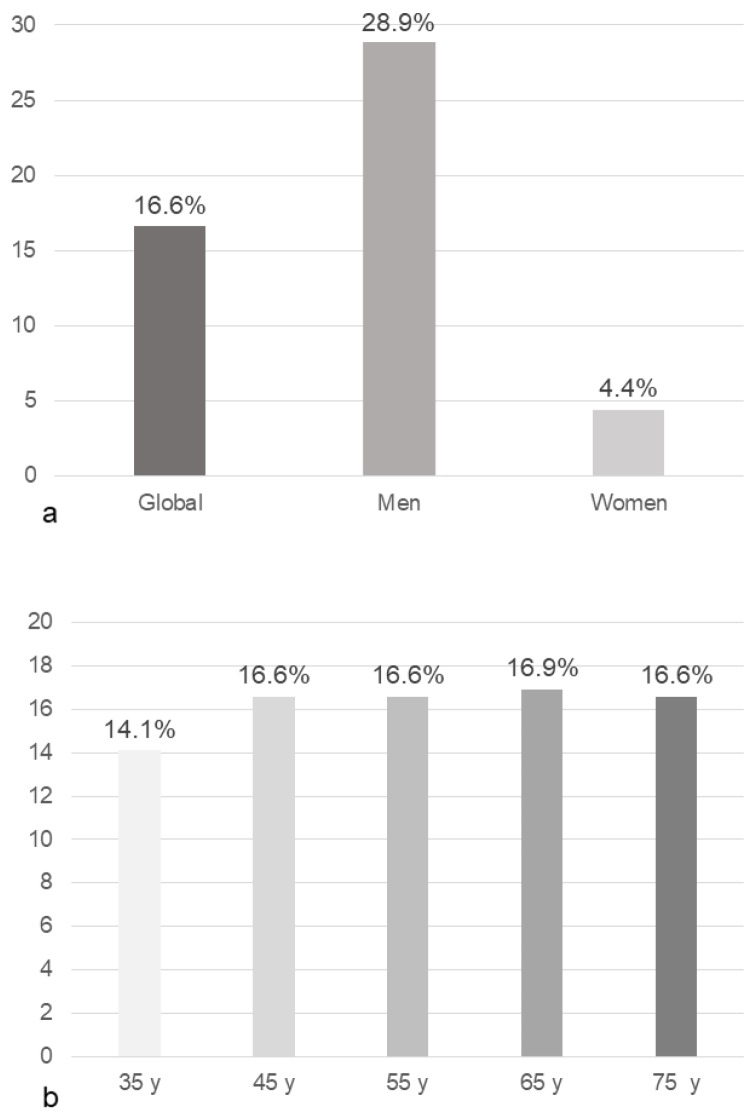
Percentage (%) of subjects with early vascular aging globally, by sex (**a**), and by age group in years (**b**). Y, years.

**Figure 2 nutrients-12-01025-f002:**
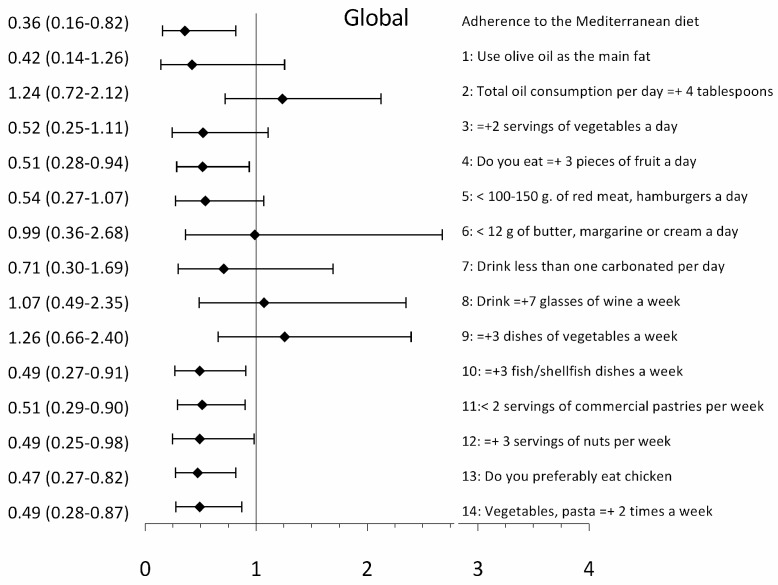
Bars show OR (odds ratio) and 95% CI. Association between arterial aging and adherence to the Mediterranean diet and its components. Dependent variable: the presence of early vascular aging (EVA) versus normal vascular aging (NVA) (1 = EVA, 0 = NVA). Independent variables: adherence to the Mediterranean diet and each of its components (1 = Yes, 0 = No). 1, For cooking, do you use olive oil as the main fat? 2, Does the total oil consumption per day ≥ 4 tablespoons? 3, Do you eat ≥ 2 servings of vegetables a day? 4, Do you eat ≥ 3 pieces of fruit a day (including natural juice)? 5, Do you eat < 100-150 g of red meat, hamburgers, sausages a day? 6, Do you eat < 12 g of butter, margarine or cream a day? 7, Do you drink less than one carbonated and/or sugary drink (soft drinks, colas, tonics, bitter) per day? 8, Do you drink ≥ 7 glasses of wine (100 ml) a week? 9, Do you eat ≥ 3 dishes of vegetables a week? (1 dish or serving of 150 g) 10, Do you eat ≥ 3 fish/shellfish dishes a week? (1 piece, dish or portion: 100–150 g of fish or 4-5 pieces or 200 g of seafood). 11, Do you eat < 2 servings of commercial (non-homemade) pastries such as cookies, custards, sweets, or cakes per week? 12, Do you eat ≥ 3 servings of nuts per week? (serving = 30 g.) 13, Do you preferably eat chicken, turkey or rabbit meat instead of beef, pork, hamburgers, or sausages? (chicken: 1 piece or serving of 100–150 g) 14, Do you eat cooked vegetables, pasta, rice, or other dishes seasoned with tomato, garlic, onion, or leek sauce cooked over low heat with olive oil (sofrito) ≥2 times a week? Adjustment variables: age (0 = <60 years, 1 = ≥60 years); sex (0 = woman; 1 = man); presence of hypertension, diabetes or obesity (0 = No; 1 = Yes); being active (0 = No; 1 = Yes), and hypotensive and hypoglycaemic drugs (0 = No; 1 = Yes). The continuous variables employed were: tobacco index, alcohol use, number of sedentary hours per week, and atherogenic index.

**Figure 3 nutrients-12-01025-f003:**
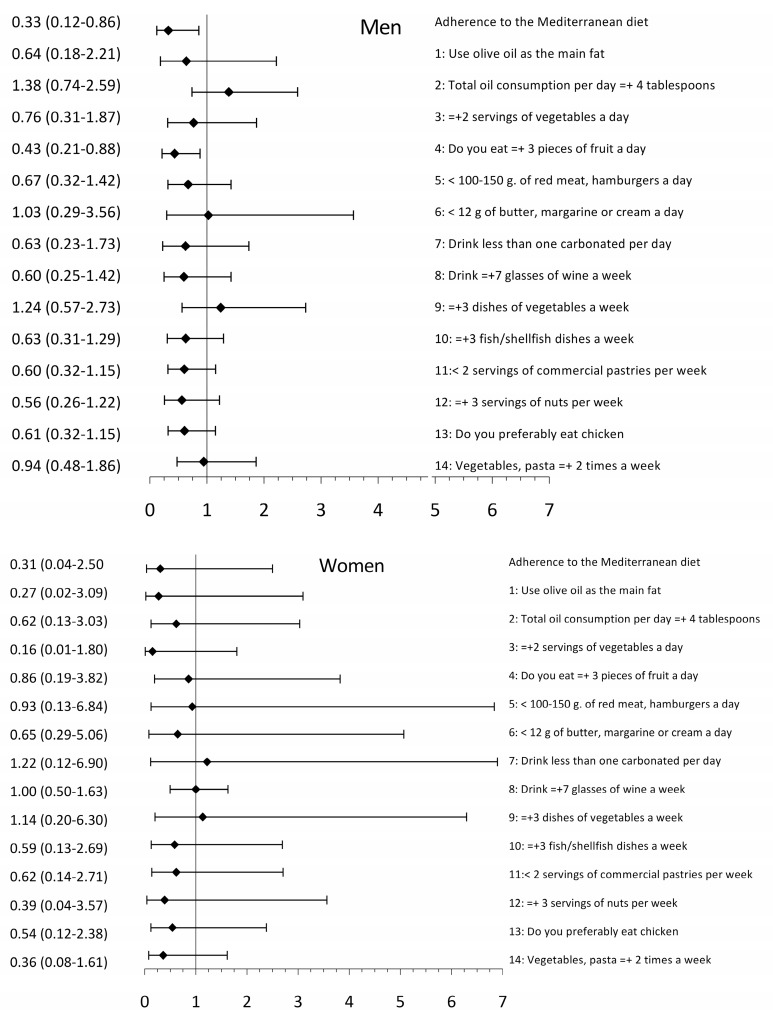
Bars show OR, (odds ratio), and 95% CI. Association between arterial aging and adherence to the Mediterranean diet and its components by sex. Dependent variable: the presence of early vascular aging (EVA) versus normal vascular aging (NVA) (1 = EVA, 0 = NVA). Independent variables: adherence to the Mediterranean diet) and each of its components (1 = Yes, 0 = No).1, For cooking, do you use olive oil as the main fat? 2, Does the total oil consumption per day ≥ 4 tablespoon? 3, Do you eat ≥ 2 servings of vegetables a day? 4, Do you eat ≥ 3 pieces of fruit a day (including natural juice)? 5, Do you eat < 100–150 g of red meat, hamburgers, sausages a day? 6, Do you eat < 12 g of butter, margarine or cream a day? 7, Do you drink less than one carbonated and/or sugary drink (soft drinks, colas, tonics, bitter) per day? 8, Do you drink ≥ 7 glasses of wine (100 ml) a week? 9, Do you eat ≥ 3 dishes of vegetables a week? (1 dish or serving of 150 g). 10, Do you eat ≥ 3 fish/shellfish dishes a week? (1 piece, dish or portion: 100–150 g of fish or 4-5 pieces or 200 g of seafood) 11, Do you eat < 2 servings of commercial (non-homemade) pastries such as cookies, custards, sweets or cakes per week? 12, Do you eat ≥ 3 servings of nuts per week? (helping = 30 g) 13, Do you preferably eat chicken, turkey or rabbit meat instead of beef, pork, hamburgers or sausages? (chicken: 1 piece or serving of 100–150 g) 14, Do you eat cooked vegetables, pasta, rice, or other dishes seasoned with tomato, garlic, onion, or leek sauce cooked over low heat with olive oil (sofrito) ≥ 2 times a week? Adjustment variables: age (0 = <60 years, 1 = ≥60 years); sex (0 = woman; 1 = man); presence of hypertension, diabetes or obesity (0 = No; 1 = Yes); being active (0 = No; 1 = Yes), and hypotensive and hypoglycaemic drugs (0 = No; 1 = Yes). The continuous variables employed were: tobacco index, alcohol use, number of sedentary hours per week and atherogenic index.

**Table 1 nutrients-12-01025-t001:** General characteristics of the subjects included globally and by vascular aging.

	Global (501)	NVA (418)	EVA (83)	*p* Value
Lifestyles				
Alcohol, (gr/W)	46.12 ± 78.25	38.89 ± 70.38	82.53 ± 102.65	<0.001
Adequate alcohol use, *n* (%)	451 (90)	383 (92)	68 (82)	0.010
Smoking index, year packages	8.88 ± 17.52	8.42 ± 17.69	11.22 ± 16.53	0.009
Smoking, *n* (%)	90 (18)	68 (14)	22 (27)	0.030
Total physical activity, (METs/ min/week)	2528 ± 1006	2439 ± 3260	2973 ± 3353	0.175
Physically Active, *n* (%)	414 (83)	340 (81)	74 (89)	0.041
Sedentary time, (h/ /week)	42.15 ± 17.57	41.22 ± 17.85	46.85 ± 16.69	0.006
Mediterranean diet	7.15 ± 2.07	7.28 ± 2.05	6.49 ± 2.09	0.002
Adherence to the MD, *n* (%)	127 (25)	115 (28)	12 (15)	0.013
Conventional risk factors				
Age, (years)	55.90 ± 14.24	55.59 ± 14.19	57.48 ± 14.44	0.270
SBP, (mmHg)	120.69 ± 3.13	118.85 ± 3.50	129.97 ± 18.65	<0.001
DBP, (mmHg)	75.53 ± 10.10	74.50 ± 9.88	80.68 ± 9.64	<0.001
Hypertension, *n* (%)	147 (29.34)	108 (26)	39 (47)	<0.001
Antihypertensive drugs, *n* (%)	96 (19)	72 (17)	24 (29)	<0.001
Total cholesterol, (mg/dl)	194.76 ± 32.50	193.69 ± 3.10	200.18 ± 28.81	0.096
LDL-cholesterol, (mg/dl)	115.51 ± 9.37	114.45 ± 9.89	120.85 ± 26.11	0.052
HDL-cholesterol, (mg/dl)	58.75 ± 16.16	59.58 ± 16.00	54.57 ± 16.41	0.010
Triglycerides, (mg/dl)	103.06 ± 53.19	98.18 ± 49.64	127.42 ± 63.13	<0.001
Atherogenic index	3.54 ± 1.07	3.41 ± 1.04	3.93 ± 1.12	<0.001
Dyslipidaemia, *n* (%)	191 (38)	75 (18)	19 (23)	0.355
Lipid–lowering drugs, *n* (%)	102 (20)	88 (21)	14 (17)	0.457
Fasting plasma glucose, (mg/dl)	88.21 ± 17.37	86.92 ± 14.73	94.70 ± 26.16	<0.001
HbA1c, (%)	5.49 ± 0.56	5.44 ± 0.47	5.74 ± 0.83	<0.001
Diabetes mellitus, *n* (%)	38 (8)	23 (5)	15 (18)	<0.001
Hypoglycaemic drugs, *n* (%)	35 (7)	21 (5)	14 (17)	0.001
Body mass index, (kg/m^2^)	26.52 ± 4.23	26.26 ± 4.23	27.81 ± 3.99	<0.001
Obesity, *n* (%)	94 (19)	75 (18)	19 (23)	<0.001
Antiplatelet drugs, *n* (%)	15 (19)	10 (2)	5 (5)	<0.041
Vascular structure and function				
cfPWV, (m/s)	6.53 ± 2.03	7.69 ± 2.00	10.76 ± 3.44	<0.001
cIMT	0.68 ± 0.11	0.67 ± 0.11	0.73 ± 0.11	<0.001

Values are means ± standard deviations for continuous data and number and proportions for categorical data. Adequate alcohol use following the recommendations in women was <140 g/week and in men <210 g/week. Adherence to the MD was considered for total scores on the 14-item Mediterranean Diet Adherence Screener (MEDAS) questionnaire of ≥9 points. Physically Active: Subjects performing at least 30 min of moderate activity 5 days a week or at least 20 min of vigorous or very vigorous activity 3 days per week were considered active. cfPWV, carotid to femoral aortic pulse wave velocity; cIMT, carotid intima media thickness; DBP, diastolic blood pressure; EVA, early vascular aging. gr/W, grams/week; h/W, hours/week; HbA1c, glycosylated hemoglobin; HDL-cholesterol, high–density lipoprotein cholesterol; LDL-cholesterol, low–density lipoprotein-cholesterol; MD, Mediterranean diet; NVA, normal vascular aging; SBP, systolic blood pressure. *p* value: differences between EVA and NVA.

**Table 2 nutrients-12-01025-t002:** Components of Mediterranean diet adherence by vascular aging.

	NVA (418)	EVA (83)	*p*
Adherence to the MD, *n* (%)	115 (23)	12 (2)	0.013
1. Do you use olive oil as the principal source of fat for cooking? *n* (%)	401 (96)	77 (93)	0.209
2. Does the total oil consumption per day ≥ 4 tablespoon? *n* (%)	156 (37)	37 (45)	0.215
3. Do you eat ≥ 2 servings of vegetables a day? *n* (%)	96 (23)	10 (12)	0.026
4. Do you eat ≥3 pieces of fruit a day (including natural juice)? *n* (%)	185 (43)	23 (28)	0.005
5. Do you eat <100–150 g of red meat, hamburgers, sausages and a day? *n* (%)	354 (85)	63 (76)	0.050
6. Do you eat <12 g of butter, margarine or cream a day? *n* (%)	387 (93)	77 (93)	0.952
7. Do you drink less than one carbonated and/or sugary drink (soft drinks, colas, tonics, bitter) per day? *n* (%)	382 (91)	74 (89)	0.512
8. Do you drink ≥ 7 glasses of wine (100 ml) a week? *n* (%)	58 (14)	26 (31)	0.001
9 Do you consume ≥ 3 dishes of vegetables a week? (1 plate or serving of 150 g), *n* (%)	74 (18)	19 (23)	0.267
10. Do you eat ≥ 3 fish-shellfish dishes a week? (1 piece, dish or portion: 100-150 g of fish or 4–5 pieces or 200 g of seafood), *n* (%)	149 (36)	23 (28)	0.081
11. Do you consume <2 servings of commercial (non-homemade) pastries such as cookies, custards, sweets or cakes per week? *n* (%)	197 (41)	28 (34)	0.025
12. Do you eat ≥ 3 servings of nuts per week? (serving = 30 g.), *n* (%)	124 (30)	14 (17)	0.017
13. Do you preferably eat chicken, turkey or rabbit meat instead of beef, pork, hamburgers or sausages? (chicken meat: 1 piece or serving of 100-150 g), *n* (%)	275 (66)	41 (49)	0.005
14. Do you eat cooked vegetables, pasta, rice, or other dishes seasoned with tomato, garlic, onion, or leek sauce cooked over low heat with olive oil (sofrito) ≥2 times a week? *n* (%)	203 (49)	27 (32)	0.007

Values are means ± standard deviations for continuous data and number and proportions for categorical data. EVA, early vascular aging. NVA, normal vascular aging. MD, Mediterranean diet. *p* value: differences between NVA and EVA.

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
