# Peer review of "Adherence to the Mediterranean Diet in Spanish Population and Its Relationship with Early Vascular Aging according to Sex and Age: EVA Study"

_nutrients, 2020, doi:10.3390/nu12041025_

Round 1
Reviewer 1 Report
In this study the Authors investigated the association between Mediterranean diet and early vascular aging. The study is duly described and the methodology is appropriate. I have only one suggestion about data presentation. In this present form, components of Mediterranean diet which decrease the risk of early vascular aging have an OR greater then 1, and vice versa. This is counterintuitive. So, I suggest to reverse the dependent variable in the logistic model to have OR below 1 when the factor is a protective one.
Author Response
Reviewer 1
Please provide a point-by-point response to the reviewer’s comments and either enter it in the box below or upload it as a Word/PDF file. Please write down "Please see the attachment." in the box if you only upload an attachment.
Response:
First of all, we want to thank you for reviewing the manuscript. In order to improve the clarity of the results, we have modified their presentation. The changes made can be seen in the following answers.
Comments and Suggestions for Authors
Point 1:
Response 1:
First of all, we want to thank you for reviewing the manuscript. In order to improve the clarity of the results, we have modified the presentation of the results. The changes made can be seen in the following answers.
Point 2: In this study the Authors investigated the association between Mediterranean diet and early vascular aging. The study is duly described and the methodology is appropriate. I have only one suggestion about data presentation. In this present form, components of Mediterranean diet which decrease the risk of early vascular aging have an OR greater then 1, and viceversa. This is counterintuitive. So, I suggest to reverse the dependent variable in the logistic model to have OR below 1 when the factor is a protective one.
Response 2:
Following the reviewer's recommendations, we have used as dependent reference variable the value (1 = EVA) in order to change the direction of the regression and obtain an OR lower than one to show a protective effect.
Moreover, taking into account the suggestions of the 2nd reviewer and the editor, we have carried out new logistic regression analysis: global and by sex (0 = woman and 1 = man).
In the current version of the manuscript, we have made the following changes:
Abstract:
Results: The adjusted logistic regression models showed that an increase in MDA decreases the probability of presenting EVA in the global analysis (OR = 0.36; 95% CI: 0.16-0.82). In the analysis by sex, the association was only maintained in men (OR = 0.33; 95% CI: 0.12-0.86), but not in women (OR = 0.31; 95% CI: 0.04-2.50). Conclusion: The results of this study suggest that a greater adherence to the Mediterranean diet decreases the probability of presenting EVA. In the analysis by sex, this association remains only in men.
Page: 2; line: 45.
2.5 Statistical Analysis
Data is presented using the mean ± standard deviation and number or percentage according to whether they are continuous or categorical variables. The comparison between NVA and EVA was performed using chi-square tests for percentages and Student's t for continuous variables. Fifteen logistic regression models were run (one for global adherence to the MDA and one for each of the 14 components), taking the presence of EVA against NVA as a dependent variable (0 = NVA, 1 = EVA) and MD and each of its components as independent variables (1 = Yes, 0 = No). To select the adjustment variables, we carried out a simple binary logistic regression analysis between the MDA and lifestyle, cardiovascular risk factors, and drugs, selecting those with values of p <0.05. As categorical variables we used: age (0 = <60 years, 1 = ≥60 years), sex (0 = woman; 1 = man); presence of hypertension, diabetes or obesity (0 = No; 1 = Yes), to be active (0 = No; 1 = Yes), and hypotensive and hypoglycaemic drugs (0 = No; 1 = Yes). The continuous variables employed were tobacco index, alcohol consumption, number of sedentary hours per week and atherogenic index. The logistic regression analysis was carried out globally and by sex. SPSS Statistics program for Windows, version 25.0 (IBM Corp, Armonk, NY) was used. We consider a value of p < 0.05 as a statistical significance limit.
Page: 5-6; line: 222-236.
Results:
3.3. Association between EVA and MDA and its components
The adjusted logistic regression models showed that an increase in the MDA decreases the probability of presenting EVA in the global analysis (OR = 0.36; 95% CI 0.16-0.82), Figure 2. In the analysis by sex, the association is only maintained in men (OR = 0.33; 95% CI 0.12-0.86), but not in women (OR = 0.31; 95% CI 0.04-2.50), Figure 3. When each of the MD components was individually analyzed in the global model, the probability of presenting EVA decreases with the consumption of: ≥ 3 pieces of fruit per day (OR = 0.51; 95% CI 0.28-0.94); ≥ 3 fish-shellfish dishes per week (OR = 0.49; 95% CI 0.27-0.91); < 2 servings of commercial (non-homemade) pastries such as cookies, custards, sweets or cakes per week (OR = 0.51; 95% CI 0.29-0.90); ≥ 3 servings of nuts per week (OR = 0.49; 95% CI 0.25-0.98); preferably eating chicken, turkey or rabbit meat instead of beef, pork, hamburgers or sausages (OR = 0.47; 95% CI 0.27-0.82); and eating cooked vegetables, pasta with tomato (OR = 0.49; 95% CI 0.25-0.98) ≥ 2 times a week, Figure 2. In the analysis by sex, the only component showing a difference was: ≥3 pieces of fruit per day decreases the probability of presenting EVA in men (OR = 0.43; 95% CI 0.21-0.88), Figure 3.
Figure 2. Association between arterial aging and adherence to the Mediterranean diet and its components. Dependent variable: the presence of early vascular aging (EVA) versus normal vascular aging (NVA) (1 = EVA, 0 = NVA). Independent variables: adherence to the Mediterranean diet (MDA) and each of its components (1 = Yes, 0 = No). MD, Mediterranean diet. MDA1, For cooking, do you use olive oil as the main fat? MDA2, Does the total oil consumption per day ≥ 4 tablespoons? MDA3, Do you eat ≥ 2 servings of vegetables a day? MDA4, Do you eat ≥ 3 pieces of fruit a day (including natural juice)? MDA5, Do you eat < 100-150 g. of red meat, hamburgers, sausages and a day? MDA6, Do you eat < 12 g of butter, margarine or cream a day? MDA7, Do you drink less than one carbonated and/or sugary drink (soft drinks, colas, tonics, bitter) per day? MDA8, Do you drink ≥ 7 glasses of wine (100 ml) a week? MDA9, Do you consume ≥ 3 dishes of vegetables a week? (1 dish or serving of 150 g.) MDA10, Do you eat ≥ 3 fish/shellfish dishes a week? (1 piece, dish or portion: 100-150 g. Of fish or 4-5 pieces or 200 g. of seafood). MDA11, Do you consume < 2 servings of commercial (non-homemade) pastries such as cookies, custards, sweets or cakes per week? MDA12, Do you eat ≥ 3 servings of nuts per week? (serving = 30 g.) MDA13, Do you preferably eat chicken, turkey or rabbit meat instead of beef, pork, hamburgers or sausages? (chicken: 1 piece or serving of 100-150 g.) MDA14, Do you consume cooked vegetables, pasta, rice, or other dishes seasoned with tomato, garlic, onion, or leek sauce cooked over low heat with olive oil (sofrito) ≥2 times a week?
Figure 3. Association between arterial aging and adherence to the Mediterranean diet and its components by sex. Dependent variable: the presence of early vascular aging (EVA) versus normal vascular aging (NVA) (1 = EVA, 0 = NVA). Independent variables: adherence to the Mediterranean diet (MDA) and each of its components (1 = Yes, 0 = No). MD, Mediterranean diet. MDA1, For cooking, do you use olive oil as the main fat? MDA1, Does the total oil consumption per day ≥ 4 tablespoon? MDA2, Does the total oil consumption per day ≥ 4 tablespoons? MDA3, Do you eat ≥ 2 servings of vegetables a day? MDA4, Do you eat ≥ 3 pieces of fruit a day (including natural juice)? MDA5, Do you eat < 100-150 g. of red meat, hamburgers, sausages and a day? MDA6, Do you eat < 12 g of butter, margarine or cream a day? MDA7, Do you drink less than one carbonated and/or sugary drink (soft drinks, colas, tonics, bitter) per day? MDA8, Do you drink ≥ 7 glasses of wine (100 ml) a week? MDA9, Do you consume ≥ 3 dishes of vegetables a week? (1 dish or serving of 150 g.). MDA10, Do you eat ≥ 3 fish/shellfish dishes a week? (1 piece, dish or portion: 100-150 g. of fish or 4-5 pieces or 200 g. of seafood) MDA11, Do you consume < 2 servings of commercial (non-homemade) pastries such as cookies, custards, sweets or cakes per week? MDA12, Do you eat ≥ 3 servings of nuts per week? (helping = 30 g.) MDA13, Do you preferably eat chicken, turkey or rabbit meat instead of beef, pork, hamburgers or sausages? (chicken: 1 piece or serving of 100-150 g.) MDA14, Do you consume cooked vegetables, pasta, rice, or other dishes seasoned with tomato, garlic, onion, or leek sauce cooked over low heat with olive oil (sofrito) ≥ 2 times a week?
Page: 10-13; line: 305-377.
Reviewer 1
Please provide a point-by-point response to the reviewer’s comments and either enter it in the box below or upload it as a Word/PDF file. Please write down "Please see the attachment." in the box if you only upload an attachment.
Response:
First of all, we want to thank you for reviewing the manuscript. In order to improve the clarity of the results, we have modified their presentation. The changes made can be seen in the following answers.
Comments and Suggestions for Authors
Point 1:
Response 1:
First of all, we want to thank you for reviewing the manuscript. In order to improve the clarity of the results, we have modified the presentation of the results. The changes made can be seen in the following answers.
Point 2: In this study the Authors investigated the association between Mediterranean diet and early vascular aging. The study is duly described and the methodology is appropriate. I have only one suggestion about data presentation. In this present form, components of Mediterranean diet which decrease the risk of early vascular aging have an OR greater then 1, and viceversa. This is counterintuitive. So, I suggest to reverse the dependent variable in the logistic model to have OR below 1 when the factor is a protective one.
Response 2:
Following the reviewer's recommendations, we have used as dependent reference variable the value (1 = EVA) in order to change the direction of the regression and obtain an OR lower than one to show a protective effect.
Moreover, taking into account the suggestions of the 2nd reviewer and the editor, we have carried out new logistic regression analysis: global and by sex (0 = woman and 1 = man).
In the current version of the manuscript, we have made the following changes:
Abstract:
Results: The adjusted logistic regression models showed that an increase in MDA decreases the probability of presenting EVA in the global analysis (OR = 0.36; 95% CI: 0.16-0.82). In the analysis by sex, the association was only maintained in men (OR = 0.33; 95% CI: 0.12-0.86), but not in women (OR = 0.31; 95% CI: 0.04-2.50). Conclusion: The results of this study suggest that a greater adherence to the Mediterranean diet decreases the probability of presenting EVA. In the analysis by sex, this association remains only in men.
Page: 2; line: 45.
2.5 Statistical Analysis
Data is presented using the mean ± standard deviation and number or percentage according to whether they are continuous or categorical variables. The comparison between NVA and EVA was performed using chi-square tests for percentages and Student's t for continuous variables. Fifteen logistic regression models were run (one for global adherence to the MDA and one for each of the 14 components), taking the presence of EVA against NVA as a dependent variable (0 = NVA, 1 = EVA) and MD and each of its components as independent variables (1 = Yes, 0 = No). To select the adjustment variables, we carried out a simple binary logistic regression analysis between the MDA and lifestyle, cardiovascular risk factors, and drugs, selecting those with values of p <0.05. As categorical variables we used: age (0 = <60 years, 1 = ≥60 years), sex (0 = woman; 1 = man); presence of hypertension, diabetes or obesity (0 = No; 1 = Yes), to be active (0 = No; 1 = Yes), and hypotensive and hypoglycaemic drugs (0 = No; 1 = Yes). The continuous variables employed were tobacco index, alcohol consumption, number of sedentary hours per week and atherogenic index. The logistic regression analysis was carried out globally and by sex. SPSS Statistics program for Windows, version 25.0 (IBM Corp, Armonk, NY) was used. We consider a value of p < 0.05 as a statistical significance limit.
Page: 5-6; line: 222-236.
Results:
3.3. Association between EVA and MDA and its components
The adjusted logistic regression models showed that an increase in the MDA decreases the probability of presenting EVA in the global analysis (OR = 0.36; 95% CI 0.16-0.82), Figure 2. In the analysis by sex, the association is only maintained in men (OR = 0.33; 95% CI 0.12-0.86), but not in women (OR = 0.31; 95% CI 0.04-2.50), Figure 3. When each of the MD components was individually analyzed in the global model, the probability of presenting EVA decreases with the consumption of: ≥ 3 pieces of fruit per day (OR = 0.51; 95% CI 0.28-0.94); ≥ 3 fish-shellfish dishes per week (OR = 0.49; 95% CI 0.27-0.91); < 2 servings of commercial (non-homemade) pastries such as cookies, custards, sweets or cakes per week (OR = 0.51; 95% CI 0.29-0.90); ≥ 3 servings of nuts per week (OR = 0.49; 95% CI 0.25-0.98); preferably eating chicken, turkey or rabbit meat instead of beef, pork, hamburgers or sausages (OR = 0.47; 95% CI 0.27-0.82); and eating cooked vegetables, pasta with tomato (OR = 0.49; 95% CI 0.25-0.98) ≥ 2 times a week, Figure 2. In the analysis by sex, the only component showing a difference was: ≥3 pieces of fruit per day decreases the probability of presenting EVA in men (OR = 0.43; 95% CI 0.21-0.88), Figure 3.
Figure 2. Association between arterial aging and adherence to the Mediterranean diet and its components. Dependent variable: the presence of early vascular aging (EVA) versus normal vascular aging (NVA) (1 = EVA, 0 = NVA). Independent variables: adherence to the Mediterranean diet (MDA) and each of its components (1 = Yes, 0 = No). MD, Mediterranean diet. MDA1, For cooking, do you use olive oil as the main fat? MDA2, Does the total oil consumption per day ≥ 4 tablespoons? MDA3, Do you eat ≥ 2 servings of vegetables a day? MDA4, Do you eat ≥ 3 pieces of fruit a day (including natural juice)? MDA5, Do you eat < 100-150 g. of red meat, hamburgers, sausages and a day? MDA6, Do you eat < 12 g of butter, margarine or cream a day? MDA7, Do you drink less than one carbonated and/or sugary drink (soft drinks, colas, tonics, bitter) per day? MDA8, Do you drink ≥ 7 glasses of wine (100 ml) a week? MDA9, Do you consume ≥ 3 dishes of vegetables a week? (1 dish or serving of 150 g.) MDA10, Do you eat ≥ 3 fish/shellfish dishes a week? (1 piece, dish or portion: 100-150 g. Of fish or 4-5 pieces or 200 g. of seafood). MDA11, Do you consume < 2 servings of commercial (non-homemade) pastries such as cookies, custards, sweets or cakes per week? MDA12, Do you eat ≥ 3 servings of nuts per week? (serving = 30 g.) MDA13, Do you preferably eat chicken, turkey or rabbit meat instead of beef, pork, hamburgers or sausages? (chicken: 1 piece or serving of 100-150 g.) MDA14, Do you consume cooked vegetables, pasta, rice, or other dishes seasoned with tomato, garlic, onion, or leek sauce cooked over low heat with olive oil (sofrito) ≥2 times a week?
Figure 3. Association between arterial aging and adherence to the Mediterranean diet and its components by sex. Dependent variable: the presence of early vascular aging (EVA) versus normal vascular aging (NVA) (1 = EVA, 0 = NVA). Independent variables: adherence to the Mediterranean diet (MDA) and each of its components (1 = Yes, 0 = No). MD, Mediterranean diet. MDA1, For cooking, do you use olive oil as the main fat? MDA1, Does the total oil consumption per day ≥ 4 tablespoon? MDA2, Does the total oil consumption per day ≥ 4 tablespoons? MDA3, Do you eat ≥ 2 servings of vegetables a day? MDA4, Do you eat ≥ 3 pieces of fruit a day (including natural juice)? MDA5, Do you eat < 100-150 g. of red meat, hamburgers, sausages and a day? MDA6, Do you eat < 12 g of butter, margarine or cream a day? MDA7, Do you drink less than one carbonated and/or sugary drink (soft drinks, colas, tonics, bitter) per day? MDA8, Do you drink ≥ 7 glasses of wine (100 ml) a week? MDA9, Do you consume ≥ 3 dishes of vegetables a week? (1 dish or serving of 150 g.). MDA10, Do you eat ≥ 3 fish/shellfish dishes a week? (1 piece, dish or portion: 100-150 g. of fish or 4-5 pieces or 200 g. of seafood) MDA11, Do you consume < 2 servings of commercial (non-homemade) pastries such as cookies, custards, sweets or cakes per week? MDA12, Do you eat ≥ 3 servings of nuts per week? (helping = 30 g.) MDA13, Do you preferably eat chicken, turkey or rabbit meat instead of beef, pork, hamburgers or sausages? (chicken: 1 piece or serving of 100-150 g.) MDA14, Do you consume cooked vegetables, pasta, rice, or other dishes seasoned with tomato, garlic, onion, or leek sauce cooked over low heat with olive oil (sofrito) ≥ 2 times a week?
Page: 10-13; line: 305-377.
Reviewer 2 Report
A definition for EVA is lacking.
Major weakness: results rely on questionnaire data and several confounders are not controlled for in the study design. The relation of EVA with MD appears an overstatement. Gender stratification appears to be the major focus of the analyses, so the aim and the title should reflect this point.
Author Response
Reviewer 2
Please provide a point-by-point response to the reviewer’s comments and either enter it in the box below or upload it as a Word/PDF file. Please write down "Please see the attachment." in the box if you only upload an attachment.
Response:
First of all, we want to thank you for reviewing the manuscript. In order to improve the clarity of the results, we have modified their presentation. The changes made can be seen in the following answers.
Comments and Suggestions for Authors
Point 1:
Response 1:
- We have edited the manuscript and we attach a certificate.
- We have reviewed the introduction and and included new references, leaving the new version as follows.
- Introduction
The Mediterranean diet (MD) consists of a high intake of vegetables and legumes, fruit, fish, white meats, pasta with tomatoes, nuts and olive oil and a low intake of red meat, dairy products and commercial pastries. It is currently considered one of the healthiest dietary patterns, with important antioxidant and anti-inflammatory effects [1]. A review of the Cochrane Library [2] showed the benefits of the Mediterranean diet in primary prevention, reducing mortality from ischemic heart disease [3], improving lipid profile, blood pressure, blood glucose and adiposity, but the level of evidence was weak-moderate and found no benefits in secondary prevention [2]. However, the relationship of the Mediterranean diet with early vascular aging (EVA) has not been studied. Only one review analyzed the beneficial effect of polyphenols (found in fruits, vegetables, greens, nuts and olive oil) on vascular aging [4].
In addition, the interest to understand the factors that influence EVA, including the MD, as well as the criteria to define it, has grown in recent years. Therefore, the search for a definition of EVA is currently being addressed by numerous authors. Thus, different definitions of EVA have also been published [5-8] in various papers using the highest cfPWV percentiles in their definitions. Recently Nilsson W et al. [9] published the vascular aging index, integrating carotid artery intima-media thickness (cIMT) and carotid-femoral pulse wave velocity (cfPWV), which reflect arterial stiffness and subclinical atherosclerosis. Finally, the association of adherence to the Mediterranean diet and its components with EVA as measured with the vascular aging index [9] has not been studied in the general Spanish population.
Given the above, the objective of this study is to analyze the influence of adherence to the Mediterranean diet and its components on EVA in a sample of the Spanish population without previous cardiovascular disease and to analyze the differences by sex.
Page: 2; line: 60-66; 73-79, 84-86.
.- We have revised the methods section, remaining in the new version as follows.
- Materials and Methods
2.1. Study Design
Descriptive transversal study of subjects recruited to investigate the association between different risk factors and early vascular aging (EVA study) [10]. Registered in ClinicalTrials.gov (NCT02623894).
Page: 3; line: 90-91.
2.2. Study Population
The EVA project was carried out in the Primary Care Research Unit (APISAL). The subjects of this study were selected using random sampling with replacement by age groups (35, 45, 55, 65 and 75 years) from an urban population. The sample comprised 501 subjects, 100 from each age group, half of each sex. Recruitment was carried out between June 2016 and November 2017. The selection of participants by age groups and sex, as well as reasons for exclusion and reference population (43,946) can be seen in the flowchart shown in Figure 1S of the supplementary material.
A detailed description of the study methodology with inclusion and exclusion criteria and response rate has been published [10, 11].
Page: 3; line: 94-96.
2.3. Ethics Approval and Consent to Participate
The study was approved on 4/5/2015 by the Salamanca ethics committee on research with medicines. The recommendations specified in the Declaration of Helsinki were followed throughout the study [12]. All subjects signed written informed consent prior to inclusion in the study.
Page: 3; line: 104-108.
2.4. Variables and measuring instruments
Before the study began, two health professionals were trained to perform the measurements and gather the completed questionnaires following a standardized protocol.
2.4.1. Assessment of carotid intima-media thickness
The cIMT was determined by two researchers of proven reliability in measurements made on 20 subjects before the beginning of the study, with intraclass and intraobserver correlation coefficients of 0.974 and 0.897 for the interobserver agreement. The device used was Sonosite Micromax® ultrasound (Sonosite Inc., Bothell, Washington, USA), with a 5-10 MHz multi-frequency high-resolution linear transducer, which automatically measures the cIMT through the Sonocal software. The cIMT measurements were performed following the protocol published by our research group [13].
Page: 3; line: 116-122.
2.4.2. The ankle arm index
The VaSera VS-1500® device (Fukuda Denshi) was used for this measurement. The presence of vascular lesion was established following the criteria included in the clinical practice guidelines [14].
Page: 4; line: 133-135.
2.4.3. Assessment of carotid-femoral pulse wave velocity (cfPWV)
The cfPWV was evaluated with the SphymgoCor System® (AtCor Medical Pty Ltd Head Office, West Ryde, Australia). All measurements were made with participants in supine position. The cfPWV, which estimates the delay with respect to the electrocardiography wave and calculates the pulse wave velocity, were analyzed. Distances were measured with a tape placed between the sternal notch and the point where the sensor was located on the carotid and femoral arteries. These were subsequently multiplied by 0.8 [15].
Page: 4; line: 140-146.
2.4.4. Definition of normal vascular aging and early vascular aging
EVA and normal vascular aging (NVA) were defined in two steps: Step 1: subjects with vascular damage in carotid arteries or peripheral artery disease were classified as EVA. Step 2: The population studied was classified by age and sex in percentiles of Vascular Aging Index (VAI) [9], which was estimated with cfPWV and intima-media thickness of the carotid artery, using the following formula VAI = (log (1.09) X 10 cIMT + log (1.14) cfPWV ) X 39.1 + 4.76. Thus, values ≥ Percentile 90 were considered EVA and those < Percentile 90 as NVA.
2.4.5. Adherence to the Mediterranean Diet
This was assessed with a 14-item questionnaire used in the PREDIMED study, previously validated in the Spanish population [16]. The questionnaire includes 12 questions about food consumption frequency and two about eating habits, with scores of zero or one. In summary, a point was awarded if subjects used olive oil as the main fat for cooking, ate more white meat than red meat, or daily consumed: more than 4 tablespoons of olive oil, 2 or more servings of vegetables, 3 or more pieces of fruit, less than 1 serving of red meat, and less than 1 sugary drink. One more point was allocated for the weekly consumption of 7 or more glasses of wine, 3 or more servings of legumes, 3 or more servings of fish, 3 or more servings of nuts or dried fruits, 2 or more servings of stir-fry, and less than 2 portions of cake or similar. The total score ranges from 0 to 14 points and MDA was considered when the sum is ≥ 9 points [16].
Page: 4; line: 163-172.
2.4.6. Evaluation of lifestyles
Smoking status was determin with a standardized questionnaire (registering whether or not the subject smoked, and if so, the number of cigarettes smoked and smoking index). We consider smokers those participants who smoked at the time of the exam or who had quit smoking during the last year. The smoking index = number of cigarettes per day / years of smoking. Alcohol consumption was measured with a standardized questionnaire (recording the type and amount of alcohol drunk during a week, measured in gr/week). Drinking free of risk was considered if alcohol drinking was less than 140 g/week for women and below 210g/week for men. Physical activity was measured with the International Physical Activity Questionnaire — Short Form (IPAQ-SF): This questionnaire consists of 9 items, includes the activity carried out by each of the participants in 4 degrees depending on the intensity: (1) intense physical activity, (2) moderate-intense activity, (3) walking and (4) being sedentary for 7 days. The data was computed in METS/min/week. Subjects performing at least 30 minutes of moderate activity 5 days a week or at least 20 minutes of vigorous or very vigorous activity 3 days per week were considered active [17]. Sedentary hours per week were evaluated by the Marshall questionnaire [18].
Page: 5; line: 188, 190 y 199.
2.4.7. Measurement of cardiovascular risk factors
The measurement of blood pressure, weight, height were carried out in the Primary Care Research Unit of Salamanca (APISAL) following the recommendations as previously published in the study protocol [10]
Subjects were considered to have hypertension if they were taking antihypertensive drugs or had blood pressure values ≥ 140/90 mmHg; to have diabetes if taking hypoglycaemic agents or had fasting plasma glucose ≥ 126 mg/dl or HbA1c ≥ 6.5%; to have dyslipidaemia if taking lipid-lowering drugs or had fasting total cholesterol ≥ 240 mg/dl, low density lipoprotein cholesterol (LDL-c) ≥ 160 mg/dl, high density lipoprotein cholesterol (HDL-c) ≤ 40 mg/dl in men and ≤ 50 mg/dl in women, or triglycerides ≥ 150 mg/dl. A value of BMI ≥ 30 kg/m2 was used to define obesity [14].
Page: 5; line: 203-205 y 220.
2.5 Statistical Analysis
Data is presented using the mean ± standard deviation and number or percentage according to whether they are continuous or categorical variables. The comparison between NVA and EVA was performed using chi-square tests for percentages and Student's t for continuous variables. Fifteen logistic regression models were run (one for global adherence to the MDA and one for each of the 14 components), taking the presence of EVA against NVA as a dependent variable (0 = NVA, 1 = EVA) and MD and each of its components as independent variables (1 = Yes, 0 = No). To select the adjustment variables, we carried out a simple binary logistic regression analysis between the MDA and lifestyle, cardiovascular risk factors, and drugs, selecting those with values of p <0.05. As categorical variables we used: age (0 = <60 years, 1 = ≥60 years), sex (0 = woman; 1 = man); presence of hypertension, diabetes or obesity (0 = No; 1 = Yes), to be active (0 = No; 1 = Yes), and hypotensive and hypoglycaemic drugs (0 = No; 1 = Yes). The continuous variables employed were tobacco index, alcohol consumption, number of sedentary hours per week and atherogenic index. The logistic regression analysis was carried out globally and by sex. SPSS Statistics program for Windows, version 25.0 (IBM Corp, Armonk, NY) was used. We consider a value of p < 0.05 as a statistical significance limit.
Page: 5-6; line: 222-236.
- We have carried out new analysis by sex and taking into account a number of possible confounding variables. Consequently, we have made new tables and figures, and we have rewritten the results section, as can be seen in the following responses.
- We have modified the conclusion. Now, after further analysis, we think that it better reflects the results found in the manuscript.
Abstract:
Conclusion: The results of this study suggest that a greater adherence to the Mediterranean diet decreases the probability of presenting EVA. In the analysis by sex, this association remains only in men.
Page: 2; line: 48-50.
- Conclusions
The results of this study suggest that a greater adherence to the Mediterranean diet decreases the probability of presenting EVA. In the analysis by sex, this association remains only in men.
Page: 14; line: 440-441.
Point 2:
A definition for EVA is lacking.
Response 2:
Indeed, as the reviewer comments, there is currently no agreed definition of EVA. However, the search for a definition of EVA is currently being addressed by numerous authors. Thus, different definitions of EVA have been published [5-8] in various papers using the highest cfPWV percentiles. For all these reasons, we believe that the definition of EVA should take into account the following variables: age, sex, cfPWV, as well as parameters that reflect the structure of the arterial wall. Therefore, we think that the vascular aging index equation published by Nilsson W et al. [9], , which integrates cIMT and cfPWV and reflects arterial stiffness and subclinical atherosclerosis, may be the criterion that better estimates vascular aging. For this reason, we consider that it is necessary to carry out works that analyze its association with different lifestyles related to EVA, such as adherence to the Mediterranean diet.
In the current version of the manuscript, the definition used in this work is reflected in the following paragraphs:
2.4.4. Definition of normal vascular aging and early vascular aging
EVA and healthy vascular aging (NVA) were defined in two steps: Step 1: subjects with vascular damage in carotid arteries or peripheral artery disease were classified as EVA. Step 2: The population studied was classified by age and sex in percentiles of Vascular Aging Index (VAI) [9], which was estimated with carotid-femoral pulse wave velocity (cfPWV) and intima-media thickness of the carotid artery, using the following formula VAI = (log (1.09) X 10 cIMT + log (1.14) cfPWV ) X 39.1 + 4.76. Thus, values ≥ Percentile 90 were considered EVA and those < Percentile 90 as NVA.
Page: 4; line: 164-160.
Point 3:
Major weakness: results rely on questionnaire data and several confounders are not controlled for in the study design. The relation of EVA with MD appears an overstatement. Gender stratification appears to be the major focus of the analyses, so the aim and the title should reflect this point.
Response 3:
- One of the limitations of the study is that it is based on a questionnaire, therefore, we have included the following sentence in the limitations section:
Thirdly, the assessment of food intake was carried out through a self-reported validated questionnaire that constitutes a subjective measurement.
Page: 14; line: 434-436.
- Control of confounding factors: Firstly, we have carried out a bibliographic review to know the effect on vascular aging of different lifestyles, cardiovascular risk factors, and drugs consumption. During the last decade, numerous published works have analyzed the factors that influence vascular aging. In summary, data from reviews and prospective studies revealed that hypertension, diabetes, obesity, smoking, as well as increased triglycerides or decreased HDL cholesterol, have a clear influence on vascular aging. Similarly, drugs used in the control and treatment of different cardiovascular risk factors seem to have a favorable effect, although it is not clear whether or not there is a difference between classes. Moreover, regular physical exercise is considered a first-line strategy to prevent aging. The relationship with diet and alcohol is not so clear [19-24]. Secondly, we have carried out a simple binary regression analysis between the MDA and the variables found in the review (lifestyle, cardiovascular risk factors, and drugs), selecting those with values of p <0.05.
We have modified the statistical analysis section, leaving it in the current version as follows:
2.5 Statistical Analysis
Data is presented using the mean ± standard deviation and number or percentage according to whether they are continuous or categorical variables. The comparison between NVA and EVA was performed using chi-square tests for percentages and Student's t for continuous variables. Fifteen logistic regression models were run (one for global adherence to the MDA and one for each of the 14 components), taking the presence of EVA against NVA as a dependent variable (0 = NVA, 1 = EVA) and MD and each of its components as independent variables (1 = Yes, 0 = No). To select the adjustment variables, we carried out a simple binary logistic regression analysis between the MDA and lifestyle, cardiovascular risk factors, and drugs, selecting those with values of p <0.05. As categorical variables we used: age (0 = <60 years, 1 = ≥60 years), sex (0 = woman; 1 = man); presence of hypertension, diabetes or obesity (0 = No; 1 = Yes), to be active (0 = No; 1 = Yes), and hypotensive and hypoglycaemic drugs (0 = No; 1 = Yes). The continuous variables employed were tobacco index, alcohol consumption, number of sedentary hours per week and atherogenic index. The logistic regression analysis was carried out globally and by sex. SPSS Statistics program for Windows, version 25.0 (IBM Corp, Armonk, NY) was used. We consider a value of p < 0.05 as a statistical significance limit.
Page: 5-6; line: 222-236.
- In addition to the global analysis, we have carried out an analysis by sex. The results suggest that there is an association between the MDA and the EVA, but when performing the analysis by sex, the association only remains in men.
As a consequence of the new analysis performed, the following changes in the manuscript were made:
Abstract:
Results: The adjusted logistic regression models showed that an increase in MDA decreases the probability of presenting EVA in the global analysis (OR = 0.36; 95% CI: 0.16-0.82). In the analysis by sex, the association was only maintained in men (OR = 0.33; 95% CI: 0.12-0.86), but not in women (OR = 0.31; 95% CI: 0.04-2.50).
Page: 2; line: 45-48.
Results:
3.3. Association between EVA and MDA and its components
The adjusted logistic regression models showed that an increase in the MDA decreases the probability of presenting EVA in the global analysis (OR = 0.36; 95% CI 0.16-0.82), Figure 2. In the analysis by sex, the association is only maintained in men (OR = 0.33; 95% CI 0.12-0.86), but not in women (OR = 0.31; 95% CI 0.04-2.50), Figure 3. When each of the MD components was individually analyzed in the global model, the probability of presenting EVA decreases with the consumption of: ≥ 3 pieces of fruit per day (OR = 0.51; 95% CI 0.28-0.94); ≥ 3 fish-shellfish dishes per week (OR = 0.49; 95% CI 0.27-0.91); < 2 servings of commercial (non-homemade) pastries such as cookies, custards, sweets or cakes per week (OR = 0.51; 95% CI 0.29-0.90); ≥ 3 servings of nuts per week (OR = 0.49; 95% CI 0.25-0.98); preferably eating chicken, turkey or rabbit meat instead of beef, pork, hamburgers or sausages (OR = 0.47; 95% CI 0.27-0.82); and eating cooked vegetables, pasta with tomato (OR = 0.49; 95% CI 0.25-0.98) ≥ 2 times a week, Figure 2. In the analysis by sex, the only component showing a difference was: ≥3 pieces of fruit per day decreases the probability of presenting EVA in men (OR = 0.43; 95% CI 0.21-0.88), Figure 3.
Figure 2. Association between arterial aging and adherence to the Mediterranean diet and its components. Dependent variable: the presence of early vascular aging (EVA) versus normal vascular aging (NVA) (1 = EVA, 0 = NVA). Independent variables: adherence to the Mediterranean diet (MDA) and each of its components (1 = Yes, 0 = No). MD, Mediterranean diet. MDA1, For cooking, do you use olive oil as the main fat? MDA2, Does the total oil consumption per day ≥ 4 tablespoons? MDA3, Do you eat ≥ 2 servings of vegetables a day? MDA4, Do you eat ≥ 3 pieces of fruit a day (including natural juice)? MDA5, Do you eat < 100-150 g. of red meat, hamburgers, sausages and a day? MDA6, Do you eat < 12 g of butter, margarine or cream a day? MDA7, Do you drink less than one carbonated and/or sugary drink (soft drinks, colas, tonics, bitter) per day? MDA8, Do you drink ≥ 7 glasses of wine (100 ml) a week? MDA9, Do you consume ≥ 3 dishes of vegetables a week? (1 dish or serving of 150 g.) MDA10, Do you eat ≥ 3 fish/shellfish dishes a week? (1 piece, dish or portion: 100-150 g. Of fish or 4-5 pieces or 200 g. of seafood). MDA11, Do you consume < 2 servings of commercial (non-homemade) pastries such as cookies, custards, sweets or cakes per week? MDA12, Do you eat ≥ 3 servings of nuts per week? (serving = 30 g.) MDA13, Do you preferably eat chicken, turkey or rabbit meat instead of beef, pork, hamburgers or sausages? (chicken: 1 piece or serving of 100-150 g.) MDA14, Do you consume cooked vegetables, pasta, rice, or other dishes seasoned with tomato, garlic, onion, or leek sauce cooked over low heat with olive oil (sofrito) ≥2 times a week?
Figure 3. Association between arterial aging and adherence to the Mediterranean diet and its components by sex. Dependent variable: the presence of early vascular aging (EVA) versus normal vascular aging (NVA) (1 = EVA, 0 = NVA). Independent variables: adherence to the Mediterranean diet (MDA) and each of its components (1 = Yes, 0 = No). MD, Mediterranean diet. MDA1, For cooking, do you use olive oil as the main fat? MDA1, Does the total oil consumption per day ≥ 4 tablespoon? MDA2, Does the total oil consumption per day ≥ 4 tablespoons? MDA3, Do you eat ≥ 2 servings of vegetables a day? MDA4, Do you eat ≥ 3 pieces of fruit a day (including natural juice)? MDA5, Do you eat < 100-150 g. of red meat, hamburgers, sausages and a day? MDA6, Do you eat < 12 g of butter, margarine or cream a day? MDA7, Do you drink less than one carbonated and/or sugary drink (soft drinks, colas, tonics, bitter) per day? MDA8, Do you drink ≥ 7 glasses of wine (100 ml) a week? MDA9, Do you consume ≥ 3 dishes of vegetables a week? (1 dish or serving of 150 g.). MDA10, Do you eat ≥ 3 fish/shellfish dishes a week? (1 piece, dish or portion: 100-150 g. of fish or 4-5 pieces or 200 g. of seafood) MDA11, Do you consume < 2 servings of commercial (non-homemade) pastries such as cookies, custards, sweets or cakes per week? MDA12, Do you eat ≥ 3 servings of nuts per week? (helping = 30 g.) MDA13, Do you preferably eat chicken, turkey or rabbit meat instead of beef, pork, hamburgers or sausages? (chicken: 1 piece or serving of 100-150 g.) MDA14, Do you consume cooked vegetables, pasta, rice, or other dishes seasoned with tomato, garlic, onion, or leek sauce cooked over low heat with olive oil (sofrito) ≥ 2 times a week?
Page: 10-13; line: 305-377.
- In order to objectively reflect the results found, we have modified the conclusion. In the current version of the manuscript it seems as follows:
Abstract:
Conclusion: The results of this study suggest that a greater adherence to the Mediterranean diet decreases the probability of presenting EVA. In the analysis by sex, this association remains only in men.
Page: 2; line: 48-50.
- Conclusions
The results of this study suggest that a greater adherence to the Mediterranean diet decreases the probability of presenting EVA. In the analysis by sex, this association remains only in men.
.- Also, we have modified the title and the objective of the manuscript in the sense that the reviewer suggests, remaining in the current version as follows
Title:
Adherence to the “Mediterranean Diet” in Spain population and Its Relationship with early vascular aging according to sex: EVA Study.
Page: 1; line: 2-4.
Objective:
Abstract:
The objective of the study is to analyze the influence of adherence to the Mediterranean diet (MDA) and its components on early vascular aging (EVA) in a Spanish population sample free of cardiovascular disease and to analyze the differences by sex.
Page: 1; line: 28-30.
Manuscript
Given the above, this study aims to analyze the influence of adherence to the Mediterranean diet and its components on EVA in a sample of the Spanish population without previous cardiovascular disease and to analyze the differences by sex.
Page: 2; line: 84-86.
- Trichopoulou A, Martinez-Gonzalez MA, Tong TY, Forouhi NG, Khandelwal S, Prabhakaran D, Mozaffarian D, de Lorgeril M: Definitions and potential health benefits of the Mediterranean diet: views from experts around the world. BMC Med 2014, 12:112.
- Rees K, Takeda A, Martin N, Ellis L, Wijesekara D, Vepa A, Das A, Hartley L, Stranges S: Mediterranean-style diet for the primary and secondary prevention of cardiovascular disease. Cochrane Database Syst Rev 2019, 3:CD009825.
- Estruch R, Ros E, Salas-Salvado J, Covas MI, Corella D, Aros F, Gomez-Gracia E, Ruiz-Gutierrez V, Fiol M, Lapetra J et al: Primary Prevention of Cardiovascular Disease with a Mediterranean Diet Supplemented with Extra-Virgin Olive Oil or Nuts. N Engl J Med 2018, 378(25):e34.
- Serino A, Salazar G: Protective Role of Polyphenols against Vascular Inflammation, Aging and Cardiovascular Disease. Nutrients 2018, 11(1).
- Cunha PG, Boutouyrie P, Nilsson PM, Laurent S: Early Vascular Ageing (EVA): Definitions and Clinical Applicability. Curr Hypertens Rev 2017, 13(1):8-15.
- Cunha PG, Cotter J, Oliveira P, Vila I, Boutouyrie P, Laurent S, Nilsson PM, Scuteri A, Sousa N: Pulse wave velocity distribution in a cohort study: from arterial stiffness to early vascular aging. J Hypertens 2015, 33(7):1438-1445.
- Botto F, Obregon S, Rubinstein F, Scuteri A, Nilsson PM, Kotliar C: Frequency of early vascular aging and associated risk factors among an adult population in Latin America: the OPTIMO study. J Hum Hypertens 2018, 32(3):219-227.
- Laurent S: Defining vascular aging and cardiovascular risk. J Hypertens 2012, 30 Suppl:S3-8.
- Nilsson Wadstrom B, Fatehali AH, Engstrom G, Nilsson PM: A Vascular Aging Index as Independent Predictor of Cardiovascular Events and Total Mortality in an Elderly Urban Population. Angiology 2019, 70(10):929-937.
- Gomez-Marcos MA, Martinez-Salgado C, Gonzalez-Sarmiento R, Hernandez-Rivas JM, Sanchez-Fernandez PL, Recio-Rodriguez JI, Rodriguez-Sanchez E, Garcia-Ortiz L: Association between different risk factors and vascular accelerated ageing (EVA study): study protocol for a cross-sectional, descriptive observational study. BMJ Open 2016, 6(6):e011031.
- Gomez-Sanchez M, Patino-Alonso MC, Gomez-Sanchez L, Recio-Rodriguez JI, Rodriguez-Sanchez E, Maderuelo-Fernandez JA, Garcia-Ortiz L, Gomez-Marcos MA: Reference values of arterial stiffness parameters and their association with cardiovascular risk factors in the Spanish population. The EVA Study. Rev Esp Cardiol (Engl Ed) 2020, 73(1):43-52.
- World Medical Association Declaration of Helsinki: ethical principles for medical research involving human subjects. JAMA 2013, 310(20):2191-2194.
- Gomez-Marcos MA, Recio-Rodriguez JI, Patino-Alonso MC, Agudo-Conde C, Gomez-Sanchez L, Gomez-Sanchez M, Rodriguez-Sanchez E, Garcia-Ortiz L: Protocol for measuring carotid intima-media thickness that best correlates with cardiovascular risk and target organ damage. Am J Hypertens 2012, 25(9):955-961.
- Williams B, Mancia G, Spiering W, Agabiti Rosei E, Azizi M, Burnier M, Clement D, Coca A, De Simone G, Dominiczak A et al: 2018 Practice Guidelines for the management of arterial hypertension of the European Society of Hypertension and the European Society of Cardiology: ESH/ESC Task Force for the Management of Arterial Hypertension. J Hypertens 2018, 36(12):2284-2309.
- Van Bortel LM, Laurent S, Boutouyrie P, Chowienczyk P, Cruickshank JK, De Backer T, Filipovsky J, Huybrechts S, Mattace-Raso FU, Protogerou AD et al: Expert consensus document on the measurement of aortic stiffness in daily practice using carotid-femoral pulse wave velocity. J Hypertens 2012, 30(3):445-448.
- Schroder H, Fito M, Estruch R, Martinez-Gonzalez MA, Corella D, Salas-Salvado J, Lamuela-Raventos R, Ros E, Salaverria I, Fiol M et al: A short screener is valid for assessing Mediterranean diet adherence among older Spanish men and women. J Nutr 2011, 141(6):1140-1145.
- Craig CL, Marshall AL, Sjostrom M, Bauman AE, Booth ML, Ainsworth BE, Pratt M, Ekelund U, Yngve A, Sallis JF et al: International physical activity questionnaire: 12-country reliability and validity. Med Sci Sports Exerc 2003, 35(8):1381-1395.
- Marshall AL, Miller YD, Burton NW, Brown WJ: Measuring total and domain-specific sitting: a study of reliability and validity. Med Sci Sports Exerc 2010, 42(6):1094-1102.
- Zhang Y, Qi L, Xu L, Sun X, Liu W, Zhou S, van de Vosse F, Greenwald SE: Effects of exercise modalities on central hemodynamics, arterial stiffness and cardiac function in cardiovascular disease: Systematic review and meta-analysis of randomized controlled trials. PLoS One 2018, 13(7):e0200829.
- Rossman MJ, LaRocca TJ, Martens CR, Seals DR: Healthy lifestyle-based approaches for successful vascular aging. J Appl Physiol (1985) 2018, 125(6):1888-1900.
- Nilsson PM, Laurent S, Cunha PG, Olsen MH, Rietzschel E, Franco OH, Ryliskyte L, Strazhesko I, Vlachopoulos C, Chen CH et al: Characteristics of healthy vascular ageing in pooled population-based cohort studies: the global Metabolic syndrome and Artery REsearch Consortium. J Hypertens 2018, 36(12):2340-2349.
- Mikael LR, Paiva AMG, Gomes MM, Sousa ALL, Jardim P, Vitorino PVO, Euzebio MB, Sousa WM, Barroso WKS: Vascular Aging and Arterial Stiffness. Arq Bras Cardiol 2017, 109(3):253-258.
- Laurent S, Boutouyrie P, Cunha PG, Lacolley P, Nilsson PM: Concept of Extremes in Vascular Aging. Hypertension 2019, 74(2):218-228.
- Ji H, Teliewubai J, Lu Y, Xiong J, Yu S, Chi C, Li J, Blacher J, Zhang Y, Xu Y: Vascular aging and preclinical target organ damage in community-dwelling elderly: the Northern Shanghai Study. J Hypertens 2018, 36(6):1391-1398.
Round 2
Reviewer 2 Report
Issues pertain the study design
Author Response
Dear Ms. Lindsey Guo
Assistant Editor of
Nutrients
Thank you for your help in reviewing this manuscript with a view to consider its publication in Nutrients.
Following the suggestions of the editor and the reviewers, we enclose a new version of the manuscript entitled: “Early vascular aging is related to lower adherence to the
Mediterranean diet in the general Spanish population: EVA Study.” (Reference number ID: nutrients-735407), together with our replies to all the issues raised.
We look forward to hearing from you. If you have any additional request or need any further information, please contact us.
Yours sincerely____________________
Manuel Angel Gomez-Marcos
Primary Care Research Unit of Salamanca (APISAL)
Av. Portugal, 37005 Salamanca, Spain
Tel: +34 923 290900 ext 53550; fax: +34 923 123644; email: magomez@usal.es
Editor Authors' Answer. General Comment:
- We would like to thank you for giving us the opportunity to revise and improve our manuscript.
- We have responded point by point to the comments.
- All the changes are marked with "Track Changes" function in Microsoft Word, citing the line number and exact change.
Comments and Suggestions for Authors of Assistant Editor
Authors tried to address the comments raised by the Reviewers and the manuscript has improved, however some ambiguities still remain.
- Although authors strongly emphasise on the importance of the MedDiet against EVA protection, looking at figures 2 and 3 I am not convinced. I wonder whether the Odds for MedDiet (0.36) in the whole sample is that important given that 6 individual components of this score show also similar protetctive effects. One would anticipate more striking effects of the MedDiet- please discuss.
Response:
We have included the following paragraph in the discussion of the manuscript.
In any case, we should remember that the effect found in this study only applies to 6 of the 14 components measured to assess adherence to MD; the global results must therefore be interpreted with caution and we should bear in mind that the effect of each of the components on EVA may differ. For all these reasons, we consider that prospective studies in the Spanish population are necessary to clarify these uncertainties. In this sense, the Amsterdam Growth and Health Longitudinal Study [1] concluded that promoting the Mediterranean diet in adolescence and early adulthood may constitute an important means of preventing arterial stiffness in adulthood.
Page: 11; line: 314-320.
- In addition these Figures need more information to be stand alone. Confounders should be mentioned as well as what figures illustrate OR? RR?
Response:
We have modified the feet of figures 2 and 3:
Figure 2: Page: 8; line: 225; 239-242.
Figure 2. Bars show OR (odds ratio) and 95% CI. Association between arterial aging and adherence to the Mediterranean diet and its components. Dependent variable: the presence of early vascular aging (EVA) versus normal vascular aging (NVA) (1 = EVA, 0 = NVA). Independent variables: adherence to the Mediterranean diet (MDA) and each of its components (1 = Yes, 0 = No). MD, Mediterranean diet. MDA1, For cooking, do you use olive oil as the main fat? MDA2, Does the total oil consumption per day ≥ 4 tablespoons? MDA3, Do you eat ≥ 2 servings of vegetables a day? MDA4, Do you eat ≥ 3 pieces of fruit a day (including natural juice)? MDA5, Do you eat < 100-150 g. of red meat, hamburgers, sausages and a day? MDA6, Do you eat < 12 g of butter, margarine or cream a day? MDA7, Do you drink less than one carbonated and/or sugary drink (soft drinks, colas, tonics, bitter) per day? MDA8, Do you drink ≥ 7 glasses of wine (100 ml) a week? MDA9, Do you eat ≥ 3 dishes of vegetables a week? (1 dish or serving of 150 g.) MDA10, Do you eat ≥ 3 fish/shellfish dishes a week? (1 piece, dish or portion: 100-150 g. Of fish or 4-5 pieces or 200 g. of seafood). MDA11, Do you eat < 2 servings of commercial (non-homemade) pastries such as cookies, custards, sweets or cakes per week? MDA12, Do you eat ≥ 3 servings of nuts per week? (serving = 30 g.) MDA13, Do you preferably eat chicken, turkey or rabbit meat instead of beef, pork, hamburgers or sausages? (chicken: 1 piece or serving of 100-150 g.) MDA14, Do you eat cooked vegetables, pasta, rice, or other dishes seasoned with tomato, garlic, onion, or leek sauce cooked over low heat with olive oil (sofrito) ≥2 times a week?. Adjustment variables: age (0 = <60 years, 1 = ≥60 years); sex (0 = woman; 1 = man); presence of hypertension, diabetes or obesity (0 = No; 1 = Yes); being active (0 = No; 1 = Yes), and hypotensive and hypoglycaemic drugs (0 = No; 1 = Yes). The continuous variables employed were: tobacco index, alcohol use, number of sedentary hours per week and atherogenic index.
Figure 3: Page: 10; line: 245; 260-264.
Figure 3. Bars show OR, (odds ratio), and 95% CI. Association between arterial aging and adherence to the Mediterranean diet and its components by sex. Dependent variable: the presence of early vascular aging (EVA) versus normal vascular aging (NVA) (1 = EVA, 0 = NVA). Independent variables: adherence to the Mediterranean diet (MDA) and each of its components (1 = Yes, 0 = No). MD, Mediterranean diet. MDA1, For cooking, do you use olive oil as the main fat? MDA1, Does the total oil consumption per day ≥ 4 tablespoon? MDA2, Does the total oil consumption per day ≥ 4 tablespoons? MDA3, Do you eat ≥ 2 servings of vegetables a day? MDA4, Do you eat ≥ 3 pieces of fruit a day (including natural juice)? MDA5, Do you eat < 100-150 g. of red meat, hamburgers, sausages and a day? MDA6, Do you eat < 12 g of butter, margarine or cream a day? MDA7, Do you drink less than one carbonated and/or sugary drink (soft drinks, colas, tonics, bitter) per day? MDA8, Do you drink ≥ 7 glasses of wine (100 ml) a week? MDA9, Do you eat ≥ 3 dishes of vegetables a week? (1 dish or serving of 150 g.). MDA10, Do you eat ≥ 3 fish/shellfish dishes a week? (1 piece, dish or portion: 100-150 g. of fish or 4-5 pieces or 200 g. of seafood) MDA11, Do you eat < 2 servings of commercial (non-homemade) pastries such as cookies, custards, sweets or cakes per week? MDA12, Do you eat ≥ 3 servings of nuts per week? (helping = 30 g.) MDA13, Do you preferably eat chicken, turkey or rabbit meat instead of beef, pork, hamburgers or sausages? (chicken: 1 piece or serving of 100-150 g.) MDA14, Do you eat cooked vegetables, pasta, rice, or other dishes seasoned with tomato, garlic, onion, or leek sauce cooked over low heat with olive oil (sofrito) ≥ 2 times a week?. Adjustment variables: age (0 = <60 years, 1 = ≥60 years); sex (0 = woman; 1 = man); presence of hypertension, diabetes or obesity (0 = No; 1 = Yes); being active (0 = No; 1 = Yes), and hypotensive and hypoglycaemic drugs (0 = No; 1 = Yes). The continuous variables employed were: tobacco index, alcohol use, number of sedentary hours per week and atherogenic index.
- Table 1:
Adequate alcohol intake is not right- You mean within the recommendations- please rephrase
In the line you mention Mediterranean diet you mena score? you should also add the score's range Adherence to the Mediterranean diet -add MEDAS score >9 in parenteses
explain under the table what physically active is.
Response:
We have addressed all the questions raised in Table 1, remaining as follows:
Page: 5; line: 183-187.
Table 1. General characteristics of the subjects included in global and by vascular aging.
|
|
Global (501) |
NVA (418) |
EVA (83) |
Pvalue |
|
Lifestyles |
|
|
|
|
|
Alcohol, (gr/W) |
46.12 ± 78.25 |
38.89±70.38 |
82.53±102.65 |
<0.001 |
|
Adequate alcohol use, n (%) |
451 (90) |
383 (92) |
68 (82) |
0.010 |
|
Smoking index, year packages |
8.88 ± 17.52 |
8.42 ± 17.69 |
11.22 ± 16.53 |
0.009 |
|
Smoking, n (%) |
90 (18) |
68 (14) |
22 (27) |
0.030 |
|
Total physical activity, (METs/ min/week) |
2528 ± 1006 |
2439 ± 3260 |
2973 ± 3353 |
0.175 |
|
Physically Active, n (%) |
414 (83) |
340 (81) |
74 (89) |
0.041 |
|
Sedentary time, (h/ /week) |
42.15 ± 17.57 |
41.22±17.85 |
46.85 ± 16.69 |
0.006 |
|
Mediterranean diet |
7.15 ± 2.07 |
7.28 ± 2.05 |
6.49 ± 2.09 |
0.002 |
|
Adherence to the MD, n (%) |
127 (25) |
115 (28) |
12 (15) |
0.013 |
|
Conventional risk factors |
|
|
|
|
|
Age, (years) |
55.90 ± 14.24 |
55.59±14.19 |
57.48 ± 14.44 |
0.270 |
|
SBP, (mmHg) |
120.69± 3.13 |
118.85±3.50 |
129.97±18.65 |
<0.001 |
|
DBP, (mmHg) |
75.53 ± 10.10 |
74.50 ± 9.88 |
80.68 ± 9.64 |
<0.001 |
|
Hypertension, n (%) |
147 (29.34) |
108 (26) |
39 (47) |
<0.001 |
|
Antihypertensive drugs, n (%) |
96 (19) |
72 (17) |
24 (29) |
<0.001 |
|
Total cholesterol, (mg/dl) |
194.76±32.50 |
193.69±3.10 |
200.18±28.81 |
0.096 |
|
LDL-cholesterol, (mg/dl) |
115.51± 9.37 |
114.45±9.89 |
120.85±26.11 |
0.052 |
|
HDL-cholesterol, (mg/dl) |
58.75 ± 16.16 |
59.58±16.00 |
54.57 ± 16.41 |
0.010 |
|
Triglycerides, (mg/dl) |
103.06±53.19 |
98.18±49.64 |
127.42±63.13 |
<0.001 |
|
Atherogenic index |
3.54 ± 1.07 |
3.41 ± 1.04 |
3.93 ± 1.12 |
<0.001 |
|
Dyslipidaemia, n (%) |
191 (38) |
75 (18) |
19 (23) |
0.355 |
|
Lipid–lowering drugs, n (%) |
102 (20) |
88 (21) |
14 (17) |
0.457 |
|
Fasting plasma glucose, (mg/dl) |
88.21 ± 17.37 |
86.92±14.73 |
94.70 ± 26.16 |
<0.001 |
|
HbA1c, (%) |
5.49 ± 0.56 |
5.44 ± 0.47 |
5.74 ± 0.83 |
<0.001 |
|
Diabetes mellitus, n (%) |
38 (8) |
23 (5) |
15 (18) |
<0.001 |
|
Hypoglycaemic drugs, n (%) |
35 (7) |
21 (5) |
14 (17) |
0.001 |
|
Body mass index, (kg/m2) |
26.52 ± 4.23 |
26.26 ± 4.23 |
27.81 ± 3.99 |
<0.001 |
|
Obesity, n (%) |
94 (19) |
75 (18) |
19 (23) |
<0.001 |
|
Antiplatelet drugs, n (%) |
15 (19) |
10 (2) |
5 (5) |
<0.041 |
|
Vascular structure and function |
|
|
|
|
|
cfPWV, (m/s) |
6.53 ± 2.03 |
7.69 ± 2.00 |
10.76 ± 3.44 |
<0.001 |
|
cIMT |
0.68 ± 0.11 |
0.67 ± 0.11 |
0.73 ± 0.11 |
<0.001 |
Values are means ± standard deviations for continuous data and number and proportions for categorical data. Adequate alcohol use following the recommendations in women was <140 g/week and in men <210 g/week. Adherence to the MD was considered for total scores on the 14-item MEDAS questionnaire of ≥ 9 points. Physically Active: Subjects performing at least 30 minutes of moderate activity 5 days a week or at least 20 minutes of vigorous or very vigorous activity 3 days per week were considered active. cfPWV, carotid to femoral aortic pulse wave velocity; cIMT, carotid intima media thickness; DBP, diastolic blood pressure; EVA, early vascular aging. gr/W, grams/week; h/W, hours/week; HbA1c, glycosylated hemoglobin; HDL-cholesterol, high–density lipoprotein cholesterol; LDL-cholesterol, low–density lipoprotein-cholesterol; MD, Mediterranean diet; NVA, normal vascular aging; SBP, systolic blood pressure. p value: differences between EVA and NVA.
- Table 3 should be renumbered as table 2- I still believe that this is not necessary and that these differences could be only mentioned in the text.
Response:
In the previous answers, following the editor's recommendations, we have removed Table 2 from the manuscript: Table 3 becomes Table 2.
We have revised the numbering, remaining in the current manuscript in the text as:
Table 2. Components of Mediterranean diet adherence by vascular aging.
Page 6; line 205.
- The whole manuscript needs serious language editing. It gives the impression that revision has been made too fast and the final version of the manuscript has many parts that need editing.
Response:
We have edited the manuscript and we attach a certificate.
References
- van de Laar RJ, Stehouwer CD, van Bussel BC, Prins MH, Twisk JW, Ferreira I: Adherence to a Mediterranean dietary pattern in early life is associated with lower arterial stiffness in adulthood: the Amsterdam Growth and Health Longitudinal Study. Journal of internal medicine 2013, 273(1):79-93.